# Exosome-Based Treatment for Atherosclerosis

**DOI:** 10.3390/ijms23021002

**Published:** 2022-01-17

**Authors:** Jeongyeon Heo, Hara Kang

**Affiliations:** 1Division of Life Sciences, College of Life Sciences and Bioengineering, Incheon National University, Incheon 22012, Korea; jyheo@inu.ac.kr; 2Institute for New Drug Development, Incheon National University, Incheon 22012, Korea

**Keywords:** atherosclerosis, exosome, intercellular communication, non-coding RNA

## Abstract

Atherosclerosis is an inflammatory disease in which lipids accumulate on the walls of blood vessels, thickening and clogging these vessels. It is well known that cell-to-cell communication is involved in the pathogenesis of atherosclerosis. Exosomes are extracellular vesicles that deliver various substances (e.g., RNA, DNA, and proteins) from the donor cell to the recipient cell and that play an important role in intercellular communication. Atherosclerosis can be either induced or inhibited through cell-to-cell communication using exosomes. An understanding of the function of exosomes as therapeutic tools and in the pathogenesis of atherosclerosis is necessary to develop new atherosclerosis therapies. In this review, we summarize the studies on the regulation of atherosclerosis through exosomes derived from multiple cells as well as research on exosome-based atherosclerosis treatment.

## 1. Introduction

Atherosclerosis is an inflammatory vascular disease caused by lipid accumulation in the vessel wall, which leads to plaque formation, narrowing the vessel lumen [1,2]. The increase of fatty substances and cholesterol in plasma due to smoking, obesity, diabetes, and vessel injury is considered to be a major factor in the development of atherosclerosis [3,4,5,6]. In the process of atherosclerosis, there are changes in the phenotype of vascular systems including endothelial cell (EC) dysfunction, vascular smooth muscle cell (VSMC) proliferation and migration, vessel calcification, inflammation, infiltration of macrophages in the plaque, and macrophage polarization [7,8,9]. These changes are not generated individually, instead they are all connected because the vessel wall consists of many types of cells including ECs, VSMCs, fibroblasts, and extracellular matrix [10]. Therefore, intercellular communication is important in atherosclerosis, as occurs for several vascular diseases.

Exosomes play an important role in cell-to-cell communication [11,12]. Exosomes are nano-sized small extracellular membranous vesicles of 30–150 nm size and are used as carriers [13,14]. Exosomes are generated from late endosomes, which in turn are formed by the inward budding of multivesicular bodies (MVBs) [15]. Exosomes can be released into extracellular spaces through exocytosis [16]. Since exosomes are derived from the endosomes of donor cells, they contain their RNA, DNA, and proteins [17,18,19,20]. Exosomes are critical mediators of intercellular communication during atherosclerosis development [21,22,23,24]. Exosomes can drive either pro-atherosclerosis inducers or vessel-protective mediators, depending on the donor cells’ condition. Exosomes derived from atherosclerosis-induced vascular or inflammatory cells transfer molecules that alter the phenotype of the recipient cell and enable atherosclerosis progression by regulating cell proliferation, migration, or inflammation. In contrast, exosomes containing molecules that have anti-atherosclerosis activities regulate recipient cells to prevent atherosclerosis (Figure 1).

Due to their biocompatibility and ability as carriers, exosomes can be used as a molecular therapeutic agent [25,26]. To use exosomes as therapeutic tools, either naive exosomes or modified exosomes carrying drugs can be used. Exosomes derived from certain types of cells have a therapeutic effect of their own as they contain diverse cargo molecules from donor cells, while modified exosomes are expected to have an added therapeutic effect as they carry other therapeutically effective substances. To prepare modified exosomes, either donor cells are transfected to overexpress siRNA, miRNA, or mRNA and exosomes are later isolated, or exosomes are directly induced to have large amounts of siRNA, miRNA, or mRNA using electroporation, lipofection, sonication, or calcium chloride treatment [27].

There are some ongoing clinical trials on the use of exosomes for the treatment of diverse diseases [28,29,30]. For example, the therapeutic efficacy of extracellular vesicles derived from mesenchymal stem cells (MSCs) in diabetes mellitus (NCT02138331), lymphoma (NCT04223622), osteoarthritis (NCT04223622), Alzheimer’s disease (NCT04388982), and ischemic stroke (NCT02458755) is currently under investigation [31,32,33]. As exosomes change the phenotype of the vascular system and regulate atherosclerosis progression, understanding the role of exosomes in atherosclerosis pathogenesis and their potential therapeutic availability for the treatment of atherosclerosis is important. In this review, we summarize studies focusing on the use of exosomes for atherosclerosis progression regulation and the type of exosomes that may be used for atherosclerosis therapy.

## 2. Exosomes That Induce Atherosclerosis

The atherosclerosis condition progresses through the communication between diverse cells using exosomes. Exosomes affect the function of VSMCs, ECs, and macrophages to regulate atherosclerosis. Understanding these exosomes and the variety of their cargo substances provides information on the progression and treatment of atherosclerosis. Several studies on this topic focus on analyzing exosomes derived from atherogenic cells induced by oxidized-low density lipoprotein (ox-LDL), diabetes inducer, or nicotine or those from atherogenic patient plasma [34,35,36,37,38,39,40]. Herein, we summarize the diversity of provenience and cargo of exosomes that promote atherosclerosis (Figure 2A).

### 2.1. Exosomes Derived from Immune Cells

Foam cell formation from macrophages induced by ox-LDL plays an important role in the early procession of atherosclerosis [41]. Foam cell-derived extracellular vesicles induce VSMC migration and adhesion. Niu et al., performed a proteomic analysis of foam cell-derived extracellular vesicles and found 269 proteins differentially expressed in these vesicles compared to those derived from normal macrophages [42]. These proteins are involved in actin cytoskeleton regulation and focal adhesion pathways, which are crucial for VSMC migration and adhesion. Therefore, foam cell-derived extracellular vesicles can induce atherosclerosis by promoting VSMC migration and adhesion.

Ox-LDL-treated THP-1 monocyte-derived exosomes induced atherosclerosis. Hu et al., observed that the expression of the long non-coding RNA (lncRNA) LIPCAR was increased in exosomes derived from ox-LDL-treated THP-1 cells [43]. LIPCAR inhibits EC proliferation by inducing apoptosis and promotes VSMC proliferation and migration by regulating CDK2 and PCNA. Indeed, LIPCAR is highly expressed in plasma from patients with atherosclerosis and is commonly associated with cardiovascular diseases [44,45]. Liu et al., found that miR-106a-3p is enriched in ox-LDL-treated THP-1 derived exosomes [46]. miR-106a-3p targets caspase9, reducing apoptosis and inducing the proliferation of VSMCs. In addition, Chen et al., found an enrichment of lncRNA growth arrest-specific 5 (GAS5) in ox-LDL-treated THP-1-derived exosomes [47]. According to previous studies, GAS5 is highly expressed in atherosclerotic plaques [48]. GAS5-overexpressing THP-1-derived exosomes induced apoptosis whereas GAS5-knockdown THP-1-derived exosomes reduced apoptosis in human umbilical vein endothelial cells (HUVECs). Therefore, ox-LDL-treated THP-1-derived exosomes carrying LIPCAR, miR-106a-3p, and GAS5 lead to atherosclerosis progression by regulating the phenotypes of ECs and VSMCs.

Martinus et al., found that exosomes derived from THP-1 cells under diabetic conditions with 25 nM glucose had high expression of HSP60, a mitochondrial molecular stress protein [49]. HSP60 induces inflammation in HUVECs by activating TLR4. Since unresolved inflammation of the endothelium in diabetes contributes to atherosclerosis [35,36], THP-1 derived exosomes in diabetic conditions may promote atherosclerosis.

Zhu et al., investigated exosomes derived from nicotine-treated macrophages as nicotine is known to induce the development of atherosclerosis [37,38,39,40]. Nicotine-treated macrophage-derived exosomes induced VSMC proliferation, migration, and formation of atherosclerotic lesions in mice. In the miRNA profile analysis, miR-21-3p was enriched in nicotine-treated macrophage-derived exosomes. miR-21-3p regulates VSMC proliferation and migration by targeting phosphatase and tensin homologs deleted on chromosome 10 (PTEN). Therefore, the authors suggested that nicotine accelerates atherosclerosis via exosomal miR-21-3p derived from macrophages.

### 2.2. Exosomes Derived from VSMC

Exosomes derived from ox-LDL-treated VSMCs promote atherosclerosis by reducing EC barrier function. According to previous studies, Krüppel-like factor 5 (KLF5) induces atherosclerosis by promoting VSMC proliferation and migration [50,51]. Zheng et al., found that KLF5 was upregulated by ox-LDL in VSMCs [52]. As KLF5 increases miR-155 levels in ox-LDL-treated VSMCs, miR-155 was enriched in exosomes released from these cells. Expression of tight junction proteins in ECs including β-catenin, vascular endothelial-cadherin, zonula occludens-1, and claudin 1 was decreased by miR-155 in ox-LDL-treated VSMC-derived exosomes, causing dysfunction of the EC barrier. Therefore, VSMC-derived exosomes under atherogenic conditions accelerate the progression of atherosclerosis by miR-155-mediated regulation of EC barrier function.

Exosomes derived from human aortic smooth muscle cells (HAoSMCs) inhibit autophagy in HUVECs. Li et al., found that miR-221/222 is enriched in HAoSMC-derived exosomes and targets PTEN, subsequently activating the Akt pathway [53]. Therefore, the enrichment of miR-221/222 in HAoSMC-derived exosomes results in the inhibition of autophagy in HUVECs, leading to the progression of atherosclerosis.

### 2.3. Exosomes Derived from EC

Exosomes released from ECs regulate VSMC phenotype changes, which play an important role in atherosclerosis [54,55,56]. LncRNA LINC01005 is highly expressed in ox-LDL-treated HUVECs, and therefore, its expression is high in the derived exosomes. This lncRNA induces the synthetic phenotype of VSMCs by promoting cell migration and proliferation [57]. LINC01005 also regulates gene expression for a synthetic phenotype by sponging miR-128-3p, which targets KLF4. As KLF4 is a negative regulator of contractile genes, the modulation of the miR-128-3p/KLF4 axis by LINC01005 in ox-LDL-treated HUVEC-derived exosomes reduces contractile markers such as α-SMA and SM22, and induces synthetic SMC markers such as OPN, which promote the synthetic phenotype of VSMCs and consequently induce atherosclerosis [58,59].

Exosomes derived from ox-LDL-treated HUVECs induce pro-inflammatory M1 polarization in THP-1 monocytes, a typical feature of atherosclerosis. Interestingly, miR-155 is highly expressed in ox-LDL-treated HUVEC-derived exosomes as well as in ox-LDL-treated VSMC-derived exosomes [52]. In ox-LDL-treated HUVECs, the expression level of KLF2, a repressor of miR-155, was decreased, resulting in an increase in both cellular and released exosomal levels of miR-155 [60]. Exosomal miR-155 regulates the progression of atherosclerosis through THP-1 pro-inflammatory M1 polarization by increasing inflammatory cytokines and M1 polarization markers such as CD80 and CD86

### 2.4. Exosomes Derived from Dendritic Cells (DCs)

Exosomes derived from mature DCs affect endothelial inflammation and atherosclerosis through the membrane tumor necrosis factor-α (TNF-α)-mediated nuclear factor (NF)-κB signaling pathway [61]. Zhong et al., reported that miR-146a was significantly increased in exosomes derived from mature DCs and affected the expression of adhesion molecules [62]. miR-146a increases the expression of VCAM-1, ICAM-1, and E-selection by targeting interleukin-1 receptor-associated kinase [63]. The modulation of these adhesion molecules induces endothelial inflammation, which leads to atherosclerosis.

## 3. Exosomes That Prevent Atherosclerosis

While exosomes derived from various cells can promote atherosclerosis, exosomes derived from MSCs, endothelial progenitor cells (EPCs), bone marrow-derived macrophages (BMDMs), and platelets have been found to ameliorate atherosclerosis. In particular, exosomal miRNAs play a role in the inhibition of atherosclerosis (Table 1). Studies on exosomes that prevent atherosclerosis can help to identify new therapeutic agents for atherosclerosis (Figure 2B). 

### 3.1. Exosomes Derived from MSC

MSC-derived exosomes have been proven to have therapeutic effects in atherosclerosis [64]. When ox-LDL-treated eosinophils were treated with exosomes derived from human umbilical cord mesenchymal stem cells (hUCMSCs), migration was inhibited, inflammatory cytokine levels were reduced, and apoptosis was promoted [64]. miR-100-5p is highly expressed in hUCMSC-derived exosomes and targets frizzled 5 (FZD5), a co-receptor of Wnt signaling. Therefore, miR-100-5p in hUCMSC-derived exosomes prevents atherosclerosis by inhibiting pro-inflammatory Wnt signaling through FZD5.

Mouse bone marrow MSC-derived exosomes prevent atherosclerosis by promoting EC proliferation due to reduced apoptosis and expression of inflammatory cytokines. miR-512-3p is enriched in MSC-derived exosomes and targets Keap1, an Nrf2 inhibitor [65]. Nrf2 is known to enhance cellular resistance to oxidative stress and consequently inhibit apoptosis in HUVECs [72]. Therefore, repression of Keap1 by miR-512-3p in MSC-derived exosomes increases Nrf2 levels, which in turn inhibits apoptosis and promotes proliferation of ECs, leading to the prevention of atherosclerosis.

In addition, MSC-derived exosomes reduced the area of atherosclerotic plaque and the infiltration of macrophages in the plaque, and promoted M2 macrophage polarization in apolipoprotein E-deficient (*ApoE*(−/−)) mice fed a high-fat diet. *ApoE*(−/−) mice fed a high-fat diet showed poor lipoprotein clearance, which promotes the development of atherosclerotic plaques due to accumulation of cholesterol ester-rich particles in the blood [73,74]. Li et al., reported that the let-7 family of miRNAs was highly expressed in MSC-derived exosomes [66]. let-7 reduces macrophage migration by targeting insulin-like growth factor 2 mRNA binding protein1 (IGF2BP1), which regulates the PTEN pathway and induces M2 polarization by targeting the high mobility group AT-hook 2 (HMGA2) responsible for regulating the NF-κB pathway. Ma et al., found that miR-21a-5p is also highly expressed in MSC-derived exosomes and that miR-21a-5p induces M2 polarization by targeting KLF6 and ERK2, whereas it reduces macrophage migration by targeting ERK2 [67]. Collectively, this further implicates MSC-derived exosomes containing miR-100-5p, miR-512-3p, let-7 family, and miR-21a-5p as therapeutic agents for atherosclerosis.

### 3.2. Exosomes Derived from EPC

Endothelial dysfunction is a well-known early marker of atherosclerosis, and the condition can be recovered by EPCs [75,76]. Bai et al., injected EPC-derived exosomes into diabetic atherosclerosis mouse models and examined the reduction in atherosclerotic plaque, oxidative stress, inflammation, and vasoconstriction, and the recovery of vasodilation [68]. The miRNA profile analysis of EPC-derived exosomes showed enrichment of certain miRNAs including miR-21a-5p, miR-222-3p, miR221-3p, miR-155-5p, and miR-29a-3p. The authors suggested that these miRNAs in EPC-derived exosomes are associated with atherosclerosis.

### 3.3. Exosomes Derived from BMDM

BMDM-derived exosomes reduced inflammation and hematopoiesis while inducing macrophage polarization in *ApoE*(−/−) mice fed a high-fat diet [69]. BMDM-derived exosomes contain miR-99a, miR-146b, and miR-378a, which have anti-inflammatory functions [77,78]. These miRNAs regulate TNF-α and NF-κB, which suppress inflammation and accelerate M2 polarization in BMDMs. Interestingly, Bouchareychas et al., found that exosomes derived from interleukin-4 (IL-4)-treated BMDMs exhibited higher levels of miR-99a, miR-146b, and miR-378a and a more potently upregulated M2 polarization by increasing the expression of *Chil3* and *Retnla* compared to normal BMDM-derived exosomes [69].

### 3.4. Exosomes Derived from Platelet

When platelets are activated, they adhere to the linear vessels and inhibit atherosclerosis [79,80]. Thrombin-activated platelet-derived exosomes have high expression levels of miR-223, miR-339, and miR-21 [70]. Exosomal miR-223 inhibits endothelial inflammation by targeting ICAM-1, leading to the prevention of atherosclerosis. Yao et al., found that thrombin-activated platelet-derived exosomal miR-25-3p also inhibits inflammation in ox-LDL-treated coronary vascular endothelial cells and in *ApoE*(−/−) mice fed a high-fat diet [71]. The miRNA miR-25-3p targets A disintegrin and metalloprotease 10 (Adam10), regulating the NF-κB signaling pathway. Therefore, thrombin-activated platelet-derived exosomes can alleviate atherosclerosis through multiple anti-inflammatory miRNAs.

## 4. Exosomes as Carriers of Therapeutic Molecules for Atherosclerosis

Exosomes are being used as therapeutic tools for atherosclerosis treatment through exploitation of their role as carriers to deliver therapeutic substances such as miRNAs, mRNA vectors, or molecules that have therapeutic ability. Some exosomes carrying specific substances have documented therapeutic effects in mouse models of atherosclerosis (Figure 3).

### 4.1. miRNA-Loaded Exosomes for Atherosclerosis Therapy

In *ApoE*(−/−) mice fed a high-fat diet, miR-125b-5p levels were lower, while its target Map4k4 levels were higher than in normal mice [81]. According to previous studies, miR-125b-5p prevents atherosclerosis by inhibiting VSMC proliferation [82]. Lin et al., isolated exosomes from mouse bone marrow-derived mesenchymal stem cells (BMSCs) transfected with miR-125b-5p and tested the feasibility of exosomes as a therapeutic tool [81]. The injection of miR-125b-5p-overexpressing BMSC-derived exosomes reduced *MCP-1* mRNA and serum levels of TNF-α and IL-6 in *ApoE*(−/−) mice fed a high-fat diet, which suppressed inflammation. The blood lipid levels and plaque formation in *ApoE*(−/−) mice fed a high-fat diet were also reduced through miR-125b-5p-mediated downregulation of Map4k4, a positive regulator of the NF-κB signaling pathway.

### 4.2. mRNA-Loaded Exosomes for Atherosclerosis Therapy

Hypercholesterolemia is a causative agent of atherosclerosis [83]. Inherited mutations in genes such as the LDL receptor (*LDLR*) can cause familial hypercholesterolemia (FH) [84]. Li et al., investigated the effects of *Ldlr* mRNA-enriched exosomes on *LDLR* deletion (*Ldlr*(−/−)) mice as an FH model [85]. *Ldlr*(−/−) mice fed a high-fat diet were injected with *Ldlr* mRNA-enriched exosomes isolated from alpha mouse liver cells (AML12) transfected with *Ldlr*-expressing vector. Injection of *Ldlr* mRNA-enriched exosomes resulted in therapeutic effects including reduced liver steatosis, inflammation, LDL levels, and atherosclerotic lesions in FH mouse models.

KLF2, whose expression is induced by shear stress, changes the endothelial cell phenotype to an atheroprotective phenotype [86]. Hergenreider et al., generated KLF2-overexpressing HUVECs by transfecting them with a lentiviral vector encoding KLF2, and injected extracellular vesicles released from these cells into *ApoE*(−/−) mice fed a high-fat diet [87]. As a result, the lesion area decreased in mice injected with extracellular vesicles derived from KLF2-overexpressing HUVECs. The authors found enrichment of miR-143 and miR-145 in extracellular vesicles derived from KLF2-overexpressing HUVECs. KLF2 transcriptionally induces the expression of miR-143 and miR-145, which are known to have an atheroprotective function [88,89,90]; the enrichment of miR-143 and miR-145 in extracellular vesicles reflects the enhanced expression levels of miR-143 and miR-145 by KLF2 overexpression in HUVECs. Therefore, extracellular vesicles derived from KLF2-overexpressing HUVECs may have a therapeutic effect on atherosclerosis through the functions of miR-143 and miR-145.

### 4.3. Drug-Loaded Exosomes for Atherosclerosis Therapy

The FDA-approved hexyl 5-aminolevulinate hydrochloride (HAL) is known to have anti-inflammatory effects by initiating the biosynthesis of anti-inflammatory bilirubin [91,92,93]. Wu et al., generated M2 macrophage-derived exosomes, showing intrinsic inflammation-tropism capability, loaded with HAL (HAL@M2 exosomes) by electroporation and tested their effect on atherosclerosis [94]. The HAL@M2 exosomes reduced the levels of inflammatory cytokines such as IL-6, TNF-α, and MMP-10 in atherosclerotic plaques of *ApoE*(−/−) mice fed a high-fat diet. When free HAL and HAL@M2 exosomes were intravenously injected into mice and their locations were tracked, most HAL@M2 exosomes were located and accumulated in the inflamed abdominal cavity, while free HAL was located mostly in the liver, indicating inflammation tropism of HAL@M2 exosomes. These results raise expectations for a more selective anti-inflammatory effect for the treatment of atherosclerosis through exosomes.

## 5. Discussion

Intercellular communication is an important process in the development of atherosclerosis. The pathogenesis of atherosclerosis does not simply progress in one direction of each cell, but instead progresses through the communication of several cell types. Substance exchange through exosomes among different cell types including ECs, VSMCs, and macrophages is known to regulate atherosclerosis progression. Phenotypes associated with atherosclerosis change according to the cells of origin of exosomes and/or recipient cells. In this review, we summarize the diverse cell-to-cell communications using exosomes that are associated with atherosclerosis. Exosomes transfer diverse proteins, miRNAs, and lncRNAs as atherosclerotic inducers or inhibitors. In addition, we provide an overview of studies showing the potential of using exosomes as a therapeutic tool for atherosclerosis.

Several early studies report the use of exosomes for the diagnosis of specific diseases [95,96]. Indeed, clinical trials have used exosomes as a diagnostic tool for cardiovascular diseases such as myocardial infarction (NCT041275941), hypertension (NCT03034265), atrial fibrillation (NCT03478410), and heart failure (NCT03837470) [30]. However, recent studies have begun to focus on the analysis of molecules within the exosomes and the use of exosomes as therapeutic tools [97,98]. Exosomes are being used as therapeutic agents to treat various diseases including acute ischemic stroke (NCT03384433) [99]. Although there is no clinical trial in atherosclerosis using exosome-based treatment, these studies are attracting attention. The therapeutic effects of a naive exosome or a modified exosome on atherosclerosis were examined in *ApoE*(−/−) mice fed a high-fat diet. MSCs, EPCs, BMDMs, and platelet-derived exosomes show therapeutic potential for atherosclerosis treatment as they transfer miRNAs. Modified exosomes carrying therapeutic agents such as miRNAs, mRNAs, or drugs also showed therapeutic effects on atherosclerosis. We noticed that enriched exosomal miRNAs identified to have therapeutic effects on atherosclerosis differ depending on the origin of the donor cell. Further investigation of miRNAs commonly enriched in exosomes that have therapeutic effects on atherosclerosis may help in the development of new therapeutic strategies using exosomes.

In atherosclerotic conditions, the expression of KLF2 and KLF5 is regulated in vascular cells, which affects the expression of certain miRNAs in exosomes derived from these cells [52,87]. Reduced KLF2 in ox-LDL-treated ECs and increased KLF5 in ox-LDL-treated VSMCs resulted in the enrichment of miR-155 in their exosomes. Since miR-155 acts as an atherosclerosis inducer, regulation of KLF2 and/or KLF5 expression in the vascular system may be important to prevent atherosclerosis. Furthermore, KLF2-overexpressed exosomes derived from EC prevented atherosclerosis by increasing the levels of atheroprotective miR-143 and miR-145 [88,89,90]. Therefore, KLF2 and KLF5 could be considered as new therapeutic targets for atherosclerosis.

Although the use of exosomes as therapeutic agents has great potential because exosomes can deliver therapeutic substances as carriers and have biocompatibility, there are still several limitations related to their temporal production and quantitative efficiency [26]. To overcome the limitation of the amount of exosomes excreted by cells, studies are in progress to create exosome mimetics [27]. For example, researchers have made efforts to create cell-derived nanosized vesicles by nanovesicle biofabrication [100,101,102]. Further studies focusing on method development for scalable production or precise targeting of exosomes may help in more efficient use of exosomes for atherosclerosis treatment.

## Figures and Tables

**Figure 1 ijms-23-01002-f001:**
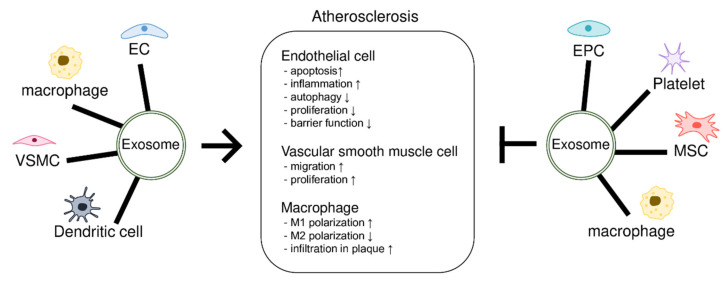
Exosome-mediated regulation of atherosclerosis. Vascular cells, macrophages, dendritic cells, endothelial progenitor cells (EPCs), and mesenchymal stem cells (MSCs) are involved in atherosclerosis through intercellular communication via exosomes. These cells release exosomes containing molecules that induce or inhibit atherosclerosis, depending on the type or physiological state of the cell. Recipient cells including endothelial cells (ECs), vascular smooth muscle cells (VSMCs), and macrophages show phenotypic changes associated with atherosclerosis. Arrow indicates induction of atherosclerosis; lines with a perpendicular line at the end indicate inhibition of atherosclerosis.

**Figure 2 ijms-23-01002-f002:**
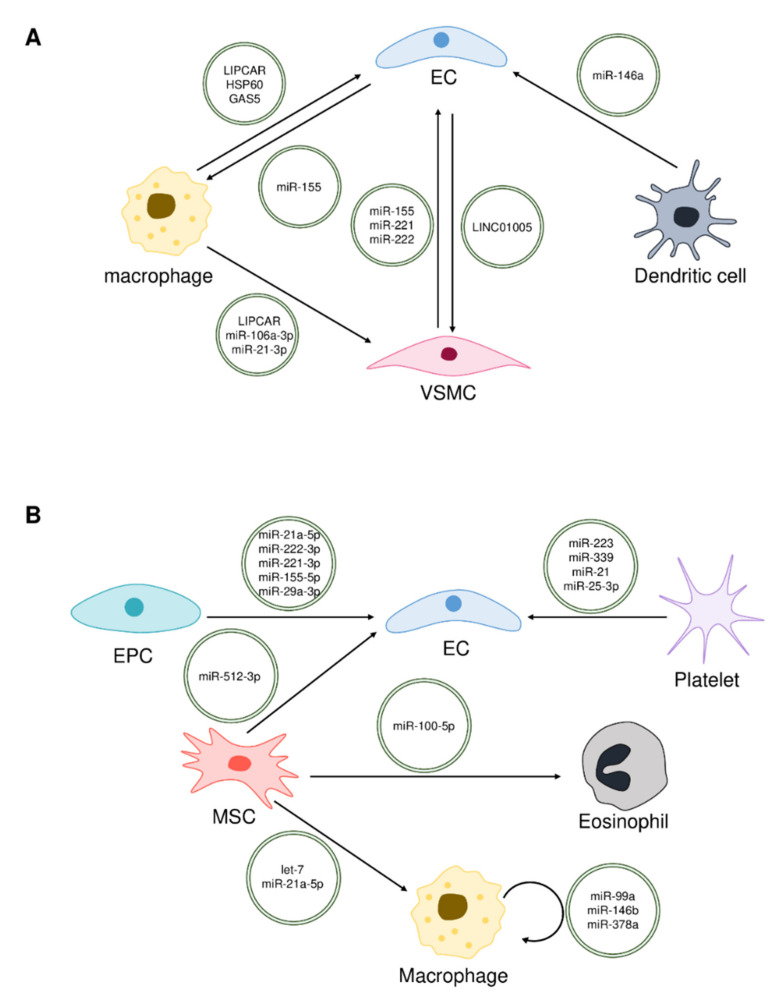
Cargos in exosomes to regulate atherosclerosis. Donor cells communicate with recipient cells via exosomes carrying various cargos, such as non-coding RNAs and proteins. Some exosomes carry cargos that induce atherosclerosis (**A**), while others carry those that inhibit atherosclerosis (**B**). The direction of the arrow indicates the transfer of exosomes from donor cells to recipient cells.

**Figure 3 ijms-23-01002-f003:**
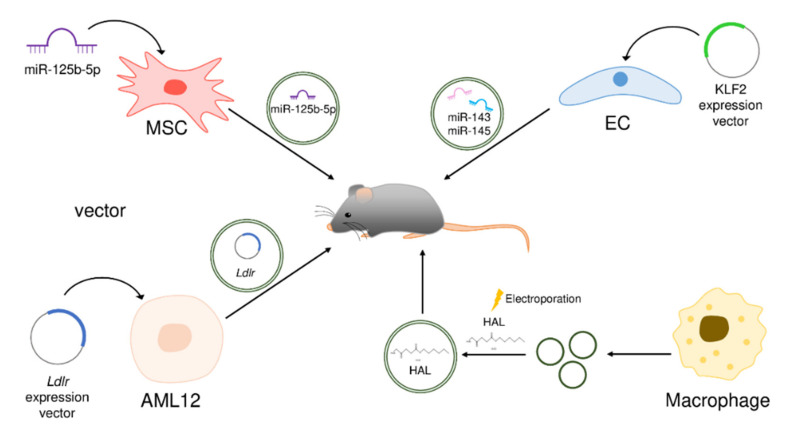
Studies on atherosclerosis therapy using exosomes.Modified exosomes carrying high levels of specific cargo were injected into atherosclerosis mouse models such as *ApoE*(−/−) or *Ldlr*(−/−) mice fed a high-fat diet by tail vein injection, and their therapeutic efficacy against atherosclerosis was investigated. Curved arrows represent transfection of cells with the indicated cargos.

**Table 1 ijms-23-01002-t001:** Exosomal miRNAs with anti-atherosclerotic effects.

Donor Cell	Recipient Cell	miRNA	Target	Phenotype	Reference
MSC	Eosinophil	miR-100-5p	FZD5	Migration ↓ Inflammation ↓ Apoptosis ↑	[64]
	EC	miR-512-3p	Keap1	Apoptosis ↓ Inflammation ↓ Proliferation ↑	[65]
	macrophage	let-7	IGF2BP1 HMGA2	Migration ↓ M2 polarization ↑	[66]
	macrophage	miR-21a-5p	ERK2 KLF6	Migration ↓ M2 polarization ↑	[67]
EPC	EC	miR-21a-5p miR-222-3p miR-221-3p miR-155-5p miR-29a-3p	N/A	Atherosclerosis plaque ↓ Oxidative stress↓ Inflammation ↓ Vasoconstriction ↓ Vasodilation ↑	[68]
BMDM	BMDM	miR-99a miR-146b miR-378a	N/A	Inflammation ↓ M2 polarization ↑	[69]
Platelet	EC	miR-223 miR-339 miR-21	ICAM-1 N/A N/A	Inflammation ↓	[70]
	EC	miR-25-3p	Adam10	Inflammation ↓	[71]

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
