# Peer review of "Exosome-Based Treatment for Atherosclerosis"

_ijms, 2022, doi:10.3390/ijms23021002_

Round 1

Reviewer 1 Report

Overall it is a good review of exosome-based treatment for atherosclerosis. A minor spell check would be useful. Also, consider using a more general term "small extracellular vesicles (sEVs)" instead of exosomes according to the International Society of Extracellular Vesicles recommendations since exosomes are considered to be a subtype of sEVs. Some previous publications claim that they isolate Exosomes but according to the method they use for isolation it is more likely a mixture of small Extracellular Vesicles. 

Author Response

Reviewer 1

Overall it is a good review of exosome-based treatment for atherosclerosis. (1) A minor spell check would be useful. (2) Also, consider using a more general term "small extracellular vesicles (sEVs)" instead of exosomes according to the International Society of Extracellular Vesicles recommendations since exosomes are considered to be a subtype of sEVs. Some previous publications claim that they isolate Exosomes but according to the method they use for isolation it is more likely a mixture of small Extracellular Vesicles. 

Response:

(1) We did a spell check. We also found and corrected mistakes in the names of the three authors in sections 2 and 3. 

(2) We agreed with the comment by the reviewer. We explained that exosomes are nano-sized small extracellular membranous vesicles of 30–150 nm size in the Introduction of the original manuscript (line 35-37). In addition, in sections 2 to 4, we denoted extracellular vesicles or exosomes according to the nomenclature of original articles.

Reviewer 2 Report

The present work presents an interesting relationship between vesicles, their contents, and atherosclerosis.

It should be emphasized that it is a very preliminary proposal, perhaps the aspect of early diagnosis rather than treatment could be emphasized.

In addition, I believe it would be useful to have a summary table with miRNAs and their possible regulatory effect, in particular on the anti-inflammatory and antioxidant action.

Author Response

Reviewer 2

The present work presents an interesting relationship between vesicles, their contents, and atherosclerosis.

  1. It should be emphasized that it is a very preliminary proposal, perhaps the aspect of early diagnosis rather than treatment could be emphasized.

Response: We agreed with the comment by the reviewer. We mentioned that recent studies have begun to focus on the use of exosomes as therapeutic tools in the discussion.

  1. In addition, I believe it would be useful to have a summary table with miRNAs and their possible regulatory effect, in particular on the anti-inflammatory and antioxidant action.

Response: We agreed with the comment by the reviewer. A summary table of exosomal miRNAs with anti-atherosclerotic effects was added (Table 1).